# Workforce Considerations When Building a Precision Medicine Program

**DOI:** 10.3390/jpm12111929

**Published:** 2022-11-19

**Authors:** Carrie L. Blout Zawatsky, Jennifer R. Leonhard, Megan Bell, Michelle M. Moore, Natasha J. Petry, Dylan M. Platt, Robert C. Green, Catherine Hajek, Kurt D. Christensen

**Affiliations:** 1Genomes2People, Department of Medicine (Genetics), Brigham and Women’s Hospital, Boston, MA 02115, USA; 2Broad Institute, Cambridge, MA 02142, USA; 3Precision Population Health, Ariadne Labs, Boston, MA 02115, USA; 4The MGH Institute of Health Professions, Boston, MA 02115, USA; 5Department of Genetics, Sanford Health, Bemidji, MN 56601, USA; 6Department of Genetics, Sanford Health, Sioux Falls, SD 57117, USA; 7Department of Genetic Counseling, Augustana University, Sioux Falls, SD 57117, USA; 8Department of Sanford Imagenetics, Sanford Health, Sioux Falls, SD 57117, USA; 9Department of Pharmacy Practice, North Dakota State University, Fargo, ND 58105, USA; 10Department of Medicine, Harvard Medical School, Boston, MA 02115, USA; 11Sanford School of Medicine, University of South Dakota, Sioux Falls, SD 57117, USA; 12Helix, San Mateo, CA 94401, USA; 13Department of Population Medicine, Harvard Medical School, Boston, MA 02215, USA; 14PRecisiOn Medicine Translational Research (PROMoTeR) Center, Department of Population Medicine, Harvard Pilgrim Health Care Institute, Boston, MA 02215, USA

**Keywords:** workforce, genetic counseling, genetic testing, pharmacogenetics, precision medicine, humans, genetic predisposition to dis-ease, delivery of health care, mass screening

## Abstract

This paper describes one healthcare system’s approach to strategically deploying genetic specialists and pharmacists to support the implementation of a precision medicine program. In 2013, Sanford Health initiated the development of a healthcare system-wide precision medicine program. Here, we report the necessary staffing including the genetic counselors, genetic counseling assistants, pharmacists, and geneticists. We examined the administrative and electronic medical records data to summarize genetic referrals over time as well as the uptake and results of an enterprise-wide genetic screening test. Between 2013 and 2020, the number of genetic specialists employed at Sanford Health increased by 190%, from 10.1 full-time equivalents (FTEs) to 29.3 FTEs. Over the same period, referrals from multiple provider types to genetic services increased by 423%, from 1438 referrals to 7517 referrals. Between 2018 and 2020, 11,771 patients received a genetic screening, with 4% identified with potential monogenic medically actionable predisposition (MAP) findings and 95% identified with at least one informative pharmacogenetic result. Of the MAP-positive patients, 85% had completed a session with a genetics provider. A strategic workforce staffing and deployment allowed Sanford Health to manage a new genetic screening program, which prompted a large increase in genetic referrals. This approach can be used as a template for other healthcare systems interested in the development of a precision medicine program.

## 1. Background

The use of genetic testing in all areas of medicine is growing, including primary care. In general, primary care providers (PCPs) view the integration of genetics into health care as inevitable, and believe genetic testing provides a clinical utility [1,2,3]. However, only a few healthcare systems have attempted to integrate proactive genetic screening into primary care [4,5,6,7,8]. One of the major barriers to a wider implementation is a lack of genetics expertise among health care providers. Approximately 25% of PCPs report that their most recent education was in medical school [9], and they often report insufficient training [10,11,12] and a lack of confidence about communicating genetic risk information to their patients [9,13]. PCPs also report concerns about anticipated workflow issues, including who is responsible for responding to results [14,15]. To fulfill the promise of genomic medicine, healthcare systems will need to be strategic about the development and support for their workforces [16,17,18,19].

Evidence suggests that PCPs can provide appropriate care and patient management [20] when adequately trained and provided access to genetic professionals. The number of genetic counselors (GCs) and other genetic specialists is already insufficient to meet the current demand for services, especially outside of academic medical centers [16,21,22]. The implementation of genetics into all aspects of care may require a paradigmatic shift in the practice of medicine by redistributing knowledge, creating alternative clinical roles, forging new professional relationships, and appropriately integrating technologies [23,24] to ensure healthcare systems properly support PCPs.

Sanford Health is one of the first healthcare systems to implement a system-wide clinical precision medicine program, including in primary care. Here, we describe how Sanford Health strategically utilized staff and other resources to support its precision medicine program and the growing demand for genetic services. This case study provides a model for healthcare systems to consider when establishing precision medicine programs.

## 2. Overview of the Healthcare System and Precision Medicine Program

Sanford Health is the nation’s largest non-profit rural healthcare system with four hubs in Bemidji, MN; Bismarck, ND; Fargo ND; and Sioux Falls, SD. In 2013, Sanford Health provided 2.1 million clinic visits with 35 hospitals and 175 clinic locations. By 2020, Sanford Health provided 5.5 million clinic visits with 46 hospitals and 210 clinic locations. Given the global focus towards precision medicine [16,25], Sanford Health launched its “Imagenetics Initiative” precision medicine program in 2014 with a vision to expand the role of genetic testing in all aspects of medicine across the entire healthcare system [26]. Sanford Health expanded its electronic medical record (EMR) system to better accommodate genetic information, automating clinical decision support (CDS) to inform medication selections and integrating genomic indicators to provide tailored genetic information and health maintenance recommendations [27]. These capabilities are maintained by committees of pharmacists and genetic specialists to assure a best practice compliance. Initial work focused on developing pharmacogenetic (PGx) systems to support *CYP2C19* genotyping to inform clopidogrel orders. Efforts culminated in 2018 with the launch of the “Sanford Chip” genetic screen (Appendix A), a genetic test available system-wide to primary care patients that offers both a preemptive PGx testing and screening for monogenic medically actionable predispositions (MAPs) [26]. The Sanford Chip genetic screen, an array-based genotyping platform, was developed to identify PGx variants with existing use guidelines, per the Clinical Pharmacogenetic Implementation Consortium (CPIC). The PGx panel started with 8 genes, and has grown to 11 genes (*CYP2C9*, *VKORC1*, *SLCO1B1*, *TPMT*, *DPYD*, *CY2C19*, *CYP3A5*, *CYP2D6*, *CYP4F2*, *CYP2C* cluster, and *IFNL3*) for 51 medications based on the CPIC guideline updates. This genetic screen also identifies potential MAPs, variants in genes associated with conditions such as hereditary cancers and heart conditions referenced by the American College of Medical Genetics and Genomics (ACMG)s V2.0 list [28] for the secondary findings’ disclosure. Details about the genetic screening platform, genes, and conditions examined have been published previously [26].

### 2.1. Personnel to Support Precision Medicine

The precision medicine program achieved its vision by coupling health care professionals with genetic expertise in both clinical and laboratory settings with PCPs and specialty providers. Each member of the clinical team plays an important part in the process of offering the genetic screen and return of results (Figure 1). Specifically, the precision medicine program hired, trained, and/or deployed genetic counseling assistants, pharmacists, laboratory genetic counselors, clinical genetic counselors and geneticists, as well as primary care providers as follows:

### 2.2. Genetic Counseling Assistants (GCAs)

In 2018, Sanford Health created a new position, “Imagenetics specialists”, with responsibilities similar to those of the genetic counseling assistants [29]. These associate or bachelor’s trained individuals work with clinical care teams and patients to inform them about genetic tests and opportunities offered through the precision medicine program. At least two specialists are located in each of the four regional hubs. Specialists serve as the primary contact for patients who want to enroll in the Sanford Biobank [30] and Sanford Chip genetic screen. They send specialized invitations that direct interested, eligible patients to the programs’ web-based consent and enrollment platforms and respond to patient questions about the process. They support clinical genetics, laboratory GCs, and pharmacists by completing administrative tasks, coordinating a sample collection, and triaging patient and provider questions and haven taken over provider education initially provided by GCs and pharmacists. In this paper, we will refer to these specialists as genetic counseling assistants (GCAs).

### 2.3. Pharmacists

Pharmacists support the precision medicine program and GCs by answering questions, serving as a reference for PGx results, and communicating with patients about results when requested to do so. Pharmacists review all PGx results and place chart notes in the EMR to alert providers when an actionable finding is identified [27]. Pharmacists may also connect with select patients to carry out medication reconciliation [31] and answer questions. Crucially, pharmacists, clinical informaticists, genetics providers, and specialty clinicians have worked to develop, test, evaluate, and refine point-of-ordering CDS that activates when PGx results indicate possible drug-gene interactions. GCs have assisted in the development of many of these alerts, including alerts surrounding genetic screening results with pharmaceutical implications. These CDS alerts are evaluated by pharmacists yearly or with each CPIC guideline update or major change in evidence, and specialty clinicians are consulted on an ad hoc basis for revisions. Any revisions of CDS are sent to the Pharmacogenomics committee for discussion and approval.

### 2.4. Physician Champions

When the initiative first launched, the four hubs selected “physician champions” who were practicing internal medicine physicians with an enthusiasm about the precision of medicine program. They provided clinical advice about the implementation as members of the precision medicine program steering committee. They were also among the first providers to introduce preemptive *CYP2C19* genotyping and the genetic screen to their patients during the pilot phases of those programs. This group of physicians also served as a resource to colleagues to answer questions. Similar to the physician champions for providers, the PGx pharmacy team identified pharmacist champions who served as resources for a PGx implementation at their individual sites.

### 2.5. Laboratory Genetic Counselors

The molecular laboratory, which employed a part-time GC before the precision medicine program launched, hired a second laboratory-based GC to support the precision medicine program. The laboratory GCs created standard operating procedures for processing the genetic screening results and contributing to the development of educational materials for providers and patients.

The laboratory GC role in the precision medicine program was originally designated to answer high-level questions from patients and providers regarding the electronic consent, the enrollment process, and the genetic test. In addition, laboratory GCs help facilitate pre- and post-test genetic counseling related to the precision medicine program as needed. In the pilot phase, the laboratory GC contacted the ordering provider to discuss the MAP-positive results to help the provider discuss the information with their patient. The process was reassessed because some patients had not reviewed their MAP-positive result with their PCP prior to their clinical genetics consult. This led to the laboratory GC notifying both the provider and the patient of the results and placing a referral for all MAP-positive patients, as well as other genetic screen patients from the precision medicine program who had questions, to see a clinical GC or geneticist. The laboratory GCs replied to patients who requested a telephone response to their questions and acted as resources for the GCAs. They also worked closely with the laboratory directors on tasks such as a variant curation and interpretation, confirmatory testing logistics, and creating MAP-positive result reports.

### 2.6. Clinical Genetics

The precision medicine program motivated the expansion of clinical genetics into primary care. Clinical GCs were selected to be available to PCPs for informal consults, same-day patient appointments, and resources for anything genetic-related. The incorporation of a clinical GC into each region increased the genetics awareness and understanding among providers and patients. GCs also played a critical role in educating providers about genetic services, as described below.

All patients who received MAP-positive results from the precision medicine program were referred to the clinical GCs or medical geneticists. Additionally, patients who had questions during enrollment, questions about uninformative results, or concerns regarding a personal or family history of a disease were referred.

### 2.7. Primary Care Providers (PCPs)

As the hub for patient care, it was critical to make PCPs aware of the precision medicine program and to empower them to respond to findings appropriately. PCPs were encouraged to engage with clinical GCs, who were embedded in primary care clinics, and were required to complete education modules designed to improve the clinician understanding of genetics [32]. Providers were also offered the opportunity to participate as patients in the precision medicine program free of charge to help provide familiarity with how the program worked and how the results were reported.

### 2.8. Provider Education and Training

From its inception, the precision medicine program engaged in a multi-faceted approach to prepare all members of clinical teams for the introduction of genetic testing, from nurses to program directors. Efforts included educational sessions and clinic in-service presentations, which were provided by GCs and pharmacists early in the initiative, and later by GCAs. Each Sanford Health family medicine, internal medicine, and obstetrics and gynecology clinic was invited to three specific educational in-service presentations about the Imagenetics Initiative over the course of two years. Sanford Health also incorporated web-based training modules about genetics into the required annual and new-provider training for all physicians and advanced practice providers [32]. Other educational resources, such as condition-specific information and frequently asked questions, were published internally in both written and video formats for providers to reference as needed. Resources were integrated whenever possible into the EMR system to facilitate providers’ abilities to distribute materials directly to their patients. There is also an ongoing communication through the intranet homepage and email newsletters.

A novel aspect of Sanford Health’s efforts was a cooperative development of its future workforce. Sanford Health worked with Augustana University to create a genetic counseling master’s program, accredited in 2015, and a genetic counseling master’s program scholarship for students contracted to be employed at Sanford Health after graduation. This graduate program has provided genetic counseling students with an exposure to precision medicine through internships, thesis projects, and volunteer opportunities with the precision medicine program. Genetic counselors from the precision medicine program serve as program faculty and provide lectures related to the precision medicine initiatives. In addition, Sanford Health developed a pharmacy training program that exposes students and residents to pharmacogenetics. With two faculty member pharmacists on the PGx team serving as primary preceptors, the PGx team allows the opportunity for a select number of fourth year pharmacy students to do five-week clinical experiences as a PGx elective in either North Dakota or South Dakota. Post-Graduate Year 1 pharmacy residents at Sanford Health in Fargo, ND, and Sioux Falls, SD, are also offered the opportunity to complete a PGx elective block. In 2021, the PGx pharmacy team welcomed their first Post-Graduate Year 2 Clinical Pharmacogenomics Resident, a training position open to a new resident every year. This program aims to create a pool of pharmacists with PGx expertise and familiarity with Sanford Health that can continue to advance the mission of the precision medicine program.

### 2.9. Data Analysis

We used administrative records and EMR data to summarize the genetic referrals, uptake, and results of the precision medicine program. The data were summarized descriptively using Microsoft^®^ Office Excel^®^ 2016 (Version 16.0, Microsoft Corporation, Redmond, WA, USA). All active departments referring to a geneticist or genetic counselor from 01 January 2013 to 31 December 2020 were reviewed. Referrals made from pediatric departments were omitted from analyses because pediatric patients were not eligible for the precision medicine program. Referrals from genetic or laboratory departments were also omitted because these referrals reflected communication orders, such as referrals to the wrong department, rather than actual referrals. The departments that placed at least one referral order to a geneticist or genetic counselor were grouped into one of the following specialty areas: oncology, surgery, family medicine, internal medicine, obstetrics/gynecology, neurology, breast clinic, cardiology, endocrinology, otolaryngology (ENT), gastroenterology, nephrology, and other. The department specialty was determined by the name of the department. Some departments in rural areas with department names based on location rather than specialty were categorized as “other”.

The protocol for the research summarized here is implemented in accordance with the Declaration of Helsinki and the ethical guidelines for medical research covering humans. The protocol was reviewed by the Sanford Institutional Review Board and determined not to be human subjects research (No. STUDY00002485/16 September 2021).

## 3. Impacts on Healthcare Utilization and Workforce

### 3.1. Genetics Service Utilization and Staffing

The number of referrals to genetics and genetic counseling services at Sanford Health increased from 1438 in 2013 to 7517 in 2020 (Figure 2). The number of departments placing referrals to genetics also increased over time from 80 in 2013 to 273 in 2020.

Anticipating the growth of genetic services, an additional 1.0 full-time equivalent (FTE) clinical GC (Table 1) was hired and positioned primarily in adult primary care clinics in each of the four regional hubs. One additional clinical geneticist FTE was also added in 2018. Clinical GC FTE increased over time, from 7 in 2013 to 22.5 in 2020. This increase in FTE included five clinical GCs personnel hired through the Augustana University genetic counseling master’s program scholarship since the Augustana University genetic counseling master’s program began.

In January 2013, clinical GCs primarily worked within maternal fetal medicine, oncology, pediatrics, neurology, and reproductive endocrinology. By December 2020, GCs had expanded into cardiology, internal medicine, and primary care settings. As the uptake of genetic screening increased, the lab GC roles added 3.0 FTE specifically for the precision medicine program to support the existing utilization management lab GC, accounting for a total of 3.6 FTEs (Table 1). Overall, between 2013 and 2020, the number of genetic specialists employed at Sanford Health increased by 190%, from 10.1 FTEs to 29.3 FTEs. Eight GCAs were also hired at 1.0 FTE each in 2018 before the genetic screen was offered and increased to 10.0 FTEs in 2020.

### 3.2. Genetic Testing and Associated Workforce Demands

Of the 11,771 patients who enrolled in the precision medicine program through 31 December 2020, 11,505 (97.7%) agreed to receive screening for MAPs (Table 2). One hundred ninety-nine (1.7%) were identified with a variant associated with an autosomal dominant disease predisposition. Another 194 (1.6%) and 96 (0.8%) individuals were found to be carriers of a variant in the *MUTYH* and *ATP7B* genes associated with autosomal recessive conditions, respectively.

A medical geneticist and/or genetic counseling session was completed for 415 of 485 MAP-positive patients (85.6%). Thirty-three of 70 MAP-positive patients who did not meet with clinical genetics (47%) had prior genetic consults, 24 (34%) were non-responsive to attempts to schedule a genetic consult, and 9 (13%) canceled or did not come for a scheduled genetic consult. Four MAP-positive patients actively declined a genetic consult.

Of the 11,680 patients from the precision medicine program with PGx data available through 31 December 2020, 11,095 (95.0%) had at least one atypical PGx finding. Of these patients, 1764 (15.9%) were identified by pharmacists to be on at least one medication for which there was an identified gene–drug interaction that warranted a further review by the prescribing provider. To handle the increased workflow for the precision medicine program, a Pharmacogenomics Manager and two clinical faculty members, each with 0.5 FTE devoted to the precision medicine program, were hired in 2018. In 2019, two full-time clinical pharmacists were added to the PGx team, and the manager position was transitioned into a director position overseeing the pharmacy, GC, and GCAs. In 2020, another 1 FTE PGx clinical pharmacist was added.

Examples illustrating how team members collaborated to addressed results from the precision medicine program, primarily through monthly patient case conferences, are summarized in Table 3. These vignettes demonstrate the various and complementary roles different providers play to ensure patient care is optimized based on findings from the precision medicine program.

## 4. Discussion

In this paper, we summarized the personnel changes that facilitated the integration of genetics into patient care across the Sanford Health system, including primary care. Sanford Health built a multi-disciplinary team that worked together to serve as a resource for PCPs and embraced clinical champions who helped to roll-out the Imagenetics Initiative. These personnel changes and proactive approaches to the workforce development, combined with provider education and EMR-based CDS, provided PCPs and other providers with the knowledge and support they needed to respond to the increased use of genetic testing. Importantly, the strategic plan Sanford Health enacted allowed the healthcare system to accommodate a large increase in genetic services with a modest increase in its genetic specialist workforce. Such plans are necessary given the expectations that shortages in the numbers of genetic specialists are likely to persist. Our results provide a unique insight into how a strategic development and the deployment of a multi-disciplinary clinical team during the implementation can empower a successful precision medicine program.

Importantly, we observed a five-fold increase in referrals to clinical genetics compared to a three-fold increase in FTEs for clinical geneticists and genetic counselors from 2013 to 2020. We also observed increases in the number of individual departments providing referrals to genetic specialists. The increases in referrals may reflect the impact of educational efforts to make providers more aware of their access to genetics specialists and feel more prepared to act on genetic information. There are external factors that also likely contributed to the growth in referrals, including the improved capabilities of genetic tests [34], an increased knowledge of the availability of genetics services among health care providers and the general public, and the publication of guidelines for using genetic information [35,36,37,38,39,40,41]. Whatever the cause, this increased referral rate suggests there were existing patient populations providers who could benefit from genetics evaluations who had not previously been referred. This is consistent with reports from other healthcare systems that have demonstrated how even patients who meet the clearly established guidelines are often not identified as at-risk or offered appropriate testing [42,43].

Our data also demonstrate one model of how healthcare systems that pursue a greater integration of genetics into patient care should be prepared to adapt staffing over time. Initially, PCPs were expected to be the first providers to discuss results with their patients, but because this follow-up was not always occurring, laboratory GCs assumed responsibilities for the initial follow-up of MAP-positive findings. It is unclear if the lack of follow-up by PCPs was due to a discomfort about discussing results, if the PCPs think this is beyond their responsibilities, or if it is a lack of time given the many competing priorities of PCPs. Regardless, this finding highlights the importance of having genetic professionals as a resource for generalists and medical subspecialists outside of genetics.

The findings also demonstrate the importance of provider education and workflows that allow genetic specialists to focus on patient care. Recognizing that genetics is a rapidly evolving area of medicine with known workforce shortages [21], Sanford Health implemented strategies to increase PCP comfort with genetic services, including mandating provider education and employing genetic counseling assistants to allow genetics clinicians to practice at the top of their scope [44,45]. These changes included delegating responsibilities for responding to basic questions about the precision medicine program and shifting educational responsibilities to genetic counseling assistants. This allowed more time for clinical GCs and pharmacists to provide consultation for patients with MAP-positive results or with more complicated questions or medical histories. Notably, our work demonstrates how many of the responsibilities of an expansion of genetics into clinical care are unlikely to be addressed solely by improvements in provider education or CDS.

Another important lesson from the Sanford Health experience, highlighted by the vignettes, was the importance of a multidisciplinary approach to providing care and evaluating patient histories and genomic screening results. In many cases, different providers’ unique perspectives contributed to improved care, including cases where multiple-provider involvement was important for a single patient. A combination of genetic results, patient history, and critical thinking led to improved care. This work suggests that when healthcare systems implement a precision medicine program, hiring and supporting a variety of specialists is essential to optimizing patient care.

## 5. Conclusions

While no single approach for the implementation of precision medicine programs is likely to be appropriate for all healthcare systems, our work highlights key points that should be addressed to increase the likelihood of success. Our experiences provide important lessons that other healthcare systems leaders can incorporate as they explore implementing their own clinical screening programs.

## Figures and Tables

**Figure 1 jpm-12-01929-f001:**
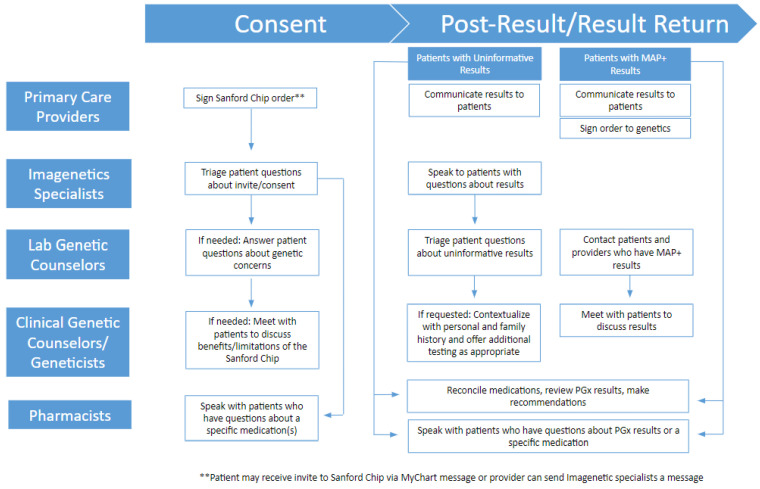
Workflow for the genetic screen through the precision medicine program. Workflow showcases the role each member of the clinical team plays in the process of offering the Sanford Chip genetic screen and return of results.

**Figure 2 jpm-12-01929-f002:**
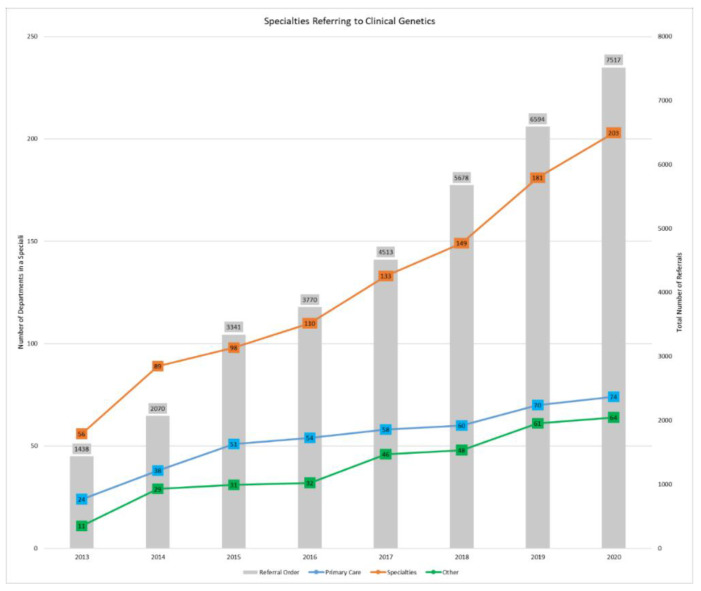
Growth over time in referring departments and clinical genetic referral orders. The line graph represents the growth in number of distinct departments referring to clinical genetics over time at Sanford Health. “Primary Care” includes family medicine and internal medicine departments. “Specialties” include obstetrics and gynecology (OB/GYN), oncology, cardiology, surgery, neurology, endocrinology, breast clinic, nephrology, gastroenterology (GI), and otolaryngology (ENT) departments. “Other” departments are defined as all other specialty departments as well as departments in rural areas with the clinic location rather than the clinic specialty listed as the department name. The bar graph represents the number of all clinical genetic referral orders not excluded from the analysis at Sanford Health from 01 January 2013 to 31 December 2020. Number of departments and number of referral orders to clinical genetics (genetic counselor and geneticist) by specialty from 01 January 2013 to 31 December 2020 are shown in Appendix A.

**Table 1 jpm-12-01929-t001:** Provider and staff changes over time in FTE for the precision medicine program.

	2013	2014	2015	2016	2017	2018	2019	2020
Genetic counseling assistants	0	0	0	0	0	8	8	10
Pharmacists	0	0	0	0	0	2	4	5
Laboratory genetic counselor	0	0	0	0	1	3.6	3.6	3.6
Clinical genetic counselor	7	10	13	13	14.9	19.9	20.5	22.5
Clinical geneticists	3.1	3.1	3	2	3.6	2.5	3.3	3.2

Total full-time equivalent (FTE) dedicated to each role by the end of each year from 2013 to 2020.

**Table 2 jpm-12-01929-t002:** Uptake of the genetic screen and genetic services through the precision medicine program over time.

	2013	2014	2015	2016	2017	2018	2019	2020	Total
Total genetic screens resulted	0	0	0	0	0	2277	5571	3923	11,771
Actionable PGx result with change in medication	0	0	0	0	0	391	671	702	1764
MAP+ genetic screens results	0	0	0	0	0	72	245	170	487
MAP+ genetic sessions completed	0	0	0	0	0	62	205	148	415

Total number of chips resulted and result outcome from the start of the program until 31 December 2020.

**Table 3 jpm-12-01929-t003:** Clinical vignettes for patients who received the genetic screen from the precision medicine program.

Program Support	Case	Outcome
Clinical Genetic Counselor	MAP+ *MSH6* pathogenic variant (Lynch syndrome)	Clinical genetic counselor reviewed the patient’s health and family history, noting the patient was in their 30s with no personal history of cancer, but had a mother with a history of uterine cancer with “weakly” focally positive pathology for *MSH6* immunohistochemistry staining that did not meet criteria for further workup at the time. Disclosure of Sanford Chip results prompted sharing of result information leading to the discovery of additional family health history consistent with Lynch syndrome. Patient and family unknowingly met NCCN criteria [33] for genetic testing. Result prompted the family to discuss family health history and realize increased risks of developing Lynch-related cancers which directly impacts recommended care for patient and at-risk family members.
Pharmacist	*CYP2D6* poor metabolizer and metoprolol (PGx response)	Pharmacist reviewed PGx findings for a patient in their 50s. The pharmacist observed the patient’s *CYP2D6 *4,*5* result indicating that he was a *CYP2D6* poor metabolizer, affecting the metabolism of an existing metoprolol order. Dutch Pharmacogenetics Working Group (DPWG) guidelines available at the time recommended decreasing the metoprolol dose by 75% or switching to an alternate agent. Patient reported shortness of breath and heart rates in the 50s and 60s while on metoprolol prompting recommendation of 75% decrease in dosage to 50 mg daily based on the result.
Laboratory Genetic Counselor	MAP uninformative	Patient in 50s called GCAs with a question regarding Sanford Chip results. GCAs transferred the patient to a lab GC. A lab counselor noted multiple family members who died of colon or related cancers, including the patient’s brother, mother, and maternal grandmother. The lab GC discussed the screening nature of the Sanford Chip and placed an order under the PCP to see a clinical GC for discussion regarding the possibility of more extensive cancer testing.
Team	MAP+ *KCNQ1* pathogenic variant (Long QT syndrome)	Patient in 60s noted to have MAP+ *KCNQ1* variant result on Sanford Chip who did not previously have a clinical diagnosis of Long QT syndrome. Diagnosis prompted EKG that showed QT interval was normal and avoidance of medications that would prolong QT was highly advised. Patient QT interval was at the upper limits of normal, so application of low-dose beta blocker was recommended. Result prompted cascade testing of at-risk family members. Patient did not have any actionable drug-gene interactions based on the PGx result, but the pharmacist noted the patient was on the muscle relaxer, tizanidine, known to prolong QT interval and not recommended in patients with Long QT syndrome. Case conference discussion led to recommending consideration of a different muscle relaxer such as cyclobenzaprine or methocarbamol as well as different medication such as gabapentin or pregabalin.

Sanford Chip patient-related scenarios describing the unique impact of various disciplines as well as team input. This table illustrates four individual patient scenarios that include: the patient’s Sanford Chip genetic result, the team member(s) involved in the patient’s care, and a brief summary of the clinical outcome after provider(s) input related to the Sanford Chip result and patient medical history.

## Data Availability

The data presented in this study are available on request from the corresponding author.

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
