# Peer review of "Workforce Considerations When Building a Precision Medicine Program"

_jpm, 2022, doi:10.3390/jpm12111929_

Round 1

Reviewer 1 Report

The manuscript of J.R. Leonhard titled Workforce Considerations when Building a Precision Medicine Program is devoted to the Precision Medicine Program at Sanford Health. The authors have convinced that their approach significantly improved the diagnostics and treatment of monogenic diseases. The manuscript is well written and I did not find any significant issues.

The minor suggestions are below:

Replace authors in brackets as [1], [2], [3], etc.

Supplement Materials should be submitted as a separate file not included in the main manuscript pdf.

Table 1 is not referred to in the manuscript text.

Table 3: Please, separate sections by lines; there are no clear separations in column 3.

Line 406: Replace Scheme 1 with Suppl. Figure 1

Line 410: Replace Scheme 1 with Suppl. Table 1

References: Where are refs # 44-48?

Line 553: Put suppl. references in the References list.

Also, I would like to know what statistical methods were used.

Author Response

Hello,

Thank you for the feedback.  Provided are your responses in black with responses from our team to address your comments in blue.

Replace authors in brackets as [1], [2], [3], etc. - Mistakenly thought https://www.mdpi.com/authors/references was including Chicago Manual Style as an option.  Updated to MDPI Reference List and Citations Style.

Supplement Materials should be submitted as a separate file not included in the main manuscript pdf. - Will upload as separate file if an option.

Table 1 is not referred to in the manuscript text. - Changed "Table 2" to "Table 1" and added reference to the missing reference for the table that is now "Table 2”.  See lines 268, 282, and 288.

Table 3: Please, separate sections by lines; there are no clear separations in column 3. - Separated sections by lines.

Line 406: Replace Scheme 1 with Suppl. Figure 1 – replaced.  See line 419

Line 410: Replace Scheme 1 with Suppl. Table 1 – replaced.  See line 421

References: Where are refs # 44-48? - Converting to MDPI Reference List and Citations Style also allowed for correction of the Reference list as the Chicago Manual Style was not number resulting in an number of errors including the References when the submission was converted by susy.mdpi.com to the MDPI Template.

Line 553: Put suppl. references in the References list. – modified.  See line 552.

Also, I would like to know what statistical methods were used. - There were no inferential statistics completed.  Based on Reviewer 2 comments, we made minor changes to the headings to make this more a case study rather than a research article.  We also have added text to be more explicit about how data were summarized descriptively.  See lines 222-223.

We greatly appreciate your time reviewing our manuscript!

Warm Regards,

Jennifer Leonhard, MS, LCGC

[email protected]

Reviewer 2 Report

This is a manuscript that describes the genetic services in Sanford Health. Although it is a description of a successful system that started in 2013, the manuscript portrays this as a study with methods and results. I believe that this valuable information can be put in a format that describes the system and the resources needed to execute precision medicine.

Figure 2 is very confusing and the legend is just a title.

Author Response

Hello,

Thank you for the feedback.  Provided are your responses in black with responses from our team to address your comments in blue.

I believe that this valuable information can be put in a format that describes the system and the resources needed to execute precision medicine. - we made minor changes to the headings to make this more a case study rather than a research article.

Figure 2 is very confusing and the legend is just a title. – There were errors when the submission was converted by susy.mdpi.com, so details were added to Figure 2 and other tables and figures.

We greatly appreciate your time reviewing our manuscript!

Warm Regards,

Jennifer Leonhard, MS, LCGC

[email protected]

Round 2

Reviewer 2 Report

This is a much improved version of the manuscript. Authors were responsive to all critiques